# Li-Ion Battery-Flywheel Hybrid Storage System: Countering Battery Aging During a Grid Frequency Regulation Service

**Sebastian Dambone Sessa \***, **Andrea Tortella**, **Mauro Andriollo** and **Roberto Benato**

Department of Industrial Engineering, University of Padova, 35131 Padova, Italy;
andrea.tortella@unipd.it (A.T.); mauro.andriollo@unipd.it (M.A.); roberto.benato@unipd.it (R.B.)
\* Correspondence: sebastian.dambonesessa@unipd.it

**Abstract:** In this paper, a hybrid storage system solution consisting of flywheels and batteries with a Lithium-manganese oxide cathode and a graphite anode is proposed, for supporting the electrical network primary frequency regulation. The aim of the paper is to investigate the benefits of flywheels in mitigation of the accelerating aging that li-ion batteries suffer during the grid frequency regulation operation. For this purpose, experimental aging tests have been performed on a lithium-manganese oxide battery module and an electrical battery model which takes into account the battery aging has been developed in a Simulink environment. Then, a flywheel electrical model has been implemented, taking into account the thermal and the electromechanical phenomena governing the electrical power exchange. This more complete model of a hybrid storage system enables us to simulate the same aging cycles of the battery-based storage system and to compare the performances of the latter with the hybrid storage system. The simulation results suggest that suitable control of the power shared between the batteries and the flywheels could effectively help in countering Li-ion battery accelerated aging due to the grid frequency regulation service.

**Keywords:** flywheel; Li-ion; battery aging; frequency regulation

## 1. Introduction

As it is widely reported in technical and scientific literature [1–8], the massive introduction of photovoltaic or wind farms in the electrical networks involves several crucial issues for the safe and reliable operation of transmission and distribution grids. Foremost among these are the non-programmability of renewable energy sources, which involves traditional power plant manage difficulties and congestions in the power transmission lines, and the decrease of the network regulating energy, which leads to network instability.

The installation of energy storage systems in a high voltage network could be one of the key elements to effectively help in solving these problems, postponing or avoiding the grid reinforcement based on new insulated cables undergrounded [9] or hosted in shared structure, [10,11] or on new overhead lines [12]. In [13], an overview of different energy storage technologies for grid services, including supercapacitor is presented. Moreover, many contributions in literature focus on the possibility to counteract the grid regulating energy decrease due to renewable energy plants by exploiting the power supplied by electrochemical energy storage installations [1–8]. This solution seems to be very effective, but recent studies demonstrated that battery aging during a grid frequency regulation operation could be higher than expected and difficult to foresee, especially for some battery technologies, including lithium-iron phosphate, lithium-nickel cobalt aluminum and lithium- nickel cobalt manganese technologies [14]. The flywheel system suitability for grid services has already been

recognised for some time [15]. Flywheels inherent features, such as the long calendar life, fast response, high round trip efficiency, high charge and discharge rates, can provide fast acting regulation without suffering in terms of life time degradation [16]. However, some issues must be likewise considered in relation to the large power sizing, such as the high investment cost, the complex power management and the integration of sophisticated technologies (flywheel rotor, electric machine, power electronics and bearings). Therefore, hybrid storage system solutions consisting of batteries and flywheels [16] could represent a valid solution to extend the battery lifetime and performances in this context. In recent literature, several contributions deal with kinetic-electrochemical hybrid systems. In [17,18], a general overview of the benefits that they could bring to the electrical system stability is presented, mainly focusing on Li-ion batteries. Paper [19] deals with the technical and economic feasibility of hybrid storage system installation in electrical networks. Paper [20] analyses a storage strategy based on lead-acid batteries and flywheels for wind power plants.

In this paper, a hybrid energy storage system consisting of flywheels and batteries with a Lithium-manganese oxide (LMO) cathode is proposed and analysed, with the aim of tackling battery aging during the grid frequency regulation operation. This goal can be reached by means of a suitable control of the power supplied by the hybrid system itself.

## 2. Battery Aging Following Frequency Regulation Cycles

The first step of this work is to investigate the aging of one of the most promising battery technologies for supporting the frequency regulation of electrical networks, i.e., a lithium-manganese oxide battery. In order to achieve this objective, accelerating aging tests have been performed by the Italian transmission system operator Terna, by cycling the battery with very fast transitions from the discharge mode to the charge one by mimicking the stress suffered by the battery during the grid frequency regulation service. In fact, in real applications, the battery unit supplied power should be controlled on the basis of the instantaneous frequency value, i.e. discharging the battery when the network frequency falls below its threshold value and vice versa. Figure 1 shows a 24-h-long frequency variation in a real network bus [14].

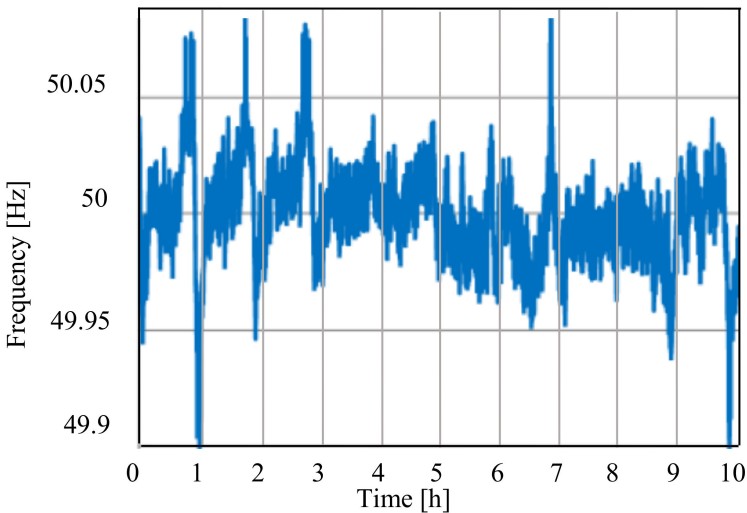

**Figure 1.** 24-h-long frequency variations in a real network bus.

The main electrical characteristics of the tested batteries are reported in Table 1.

**Table 1.** Electrical characteristics of the tested technology.

| Rank Characteristics | | Module Characteristics | |
| --- | --- | --- | --- |
| Number of modules | 16 module + 1 BMS | Number of cells | 16 + 1 BMS |
| Dimensions | 1049 × 549 × 1851 mm | Dimensions | 485 × 510 × 162.5 mm |
| Weight | 1000 kg | Weight | 48 kg |
| Capacity | 60 Ah | Capacity | 60 Ah |
| Rated voltage | 947.2 V | Rated voltage | 59.2 V |
| Nominal Energy | 47.7 KWh | Nominal Energy | 2.98 kWh |
| Operating voltage range | 768–1054.7 V | Operating voltage range | 48–65.92 V |
| Rated discharge time | 1 h | Rated discharge time | 1 h |

The aging test was carried out on a 3-kW battery module by performing 500 very stressful cycles, which lasted 10.2 h each. At the end of each cycle, the battery was completely recharged. In Figure 2 the voltage and current (the current profile follows a typical grid required power profile during the frequency regulation operation) measured during two aging cycles are reported and the recharge operations are highlighted.

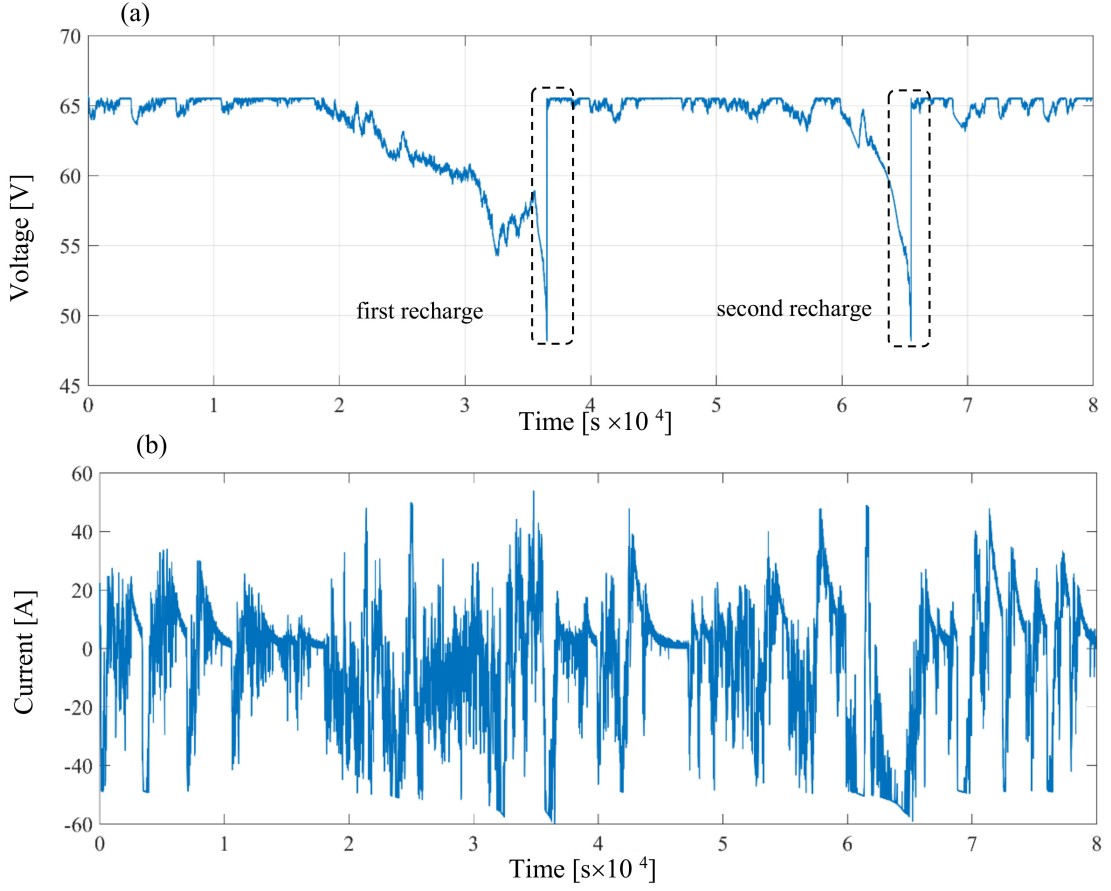

**Figure 2.** Measured voltage (**a**) and current (**b**) during two aging cycles.

Once all the aging cycles were performed, the battery capacity was reduced by 7.5%. Figure 3 shows the battery capacity decrease during the aging test on the basis of the cycles. The anomalous capacity rise during the test, which is visible in Figure 3, is chiefly due to the fact that the battery State of Charge (SoC) is calculated by means of the battery management system (BMS), starting from the integral of the charge/discharge current and by subsequently comparing the current integral with the battery capacity. Each BMS needs to be periodically re-calibrated to correct the current integration

errors but during the tests no BMS re-calibration was performed, except in case of anomalous behaviour of the batteries.

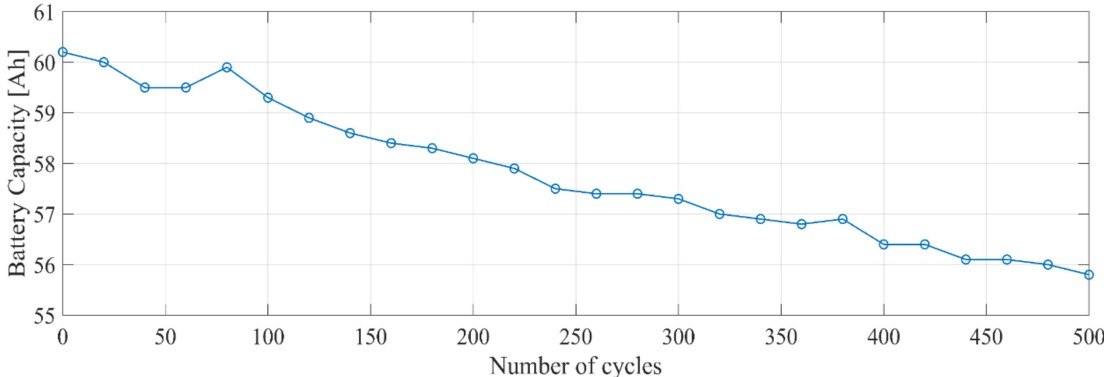

**Figure 3.** Battery capacity decrease during the aging test.

In order to identify a proper strategy to limit this aging effect, an electrical model of a battery module was developed in the Matlab Simulink environment, by using the Thevenin second order dynamic modelling approach described in [21], whose circuital representation is shown in Figure 4. The model parameters are inferred from specific voltage experimental measurements. In particular, the computation of the equivalent parameters is performed by exploiting the battery relaxation period after a discharge (see Figure 4, where the discharge phase starts at 125 s) at different battery SoCs, temperatures $\vartheta$ and currents $i$.

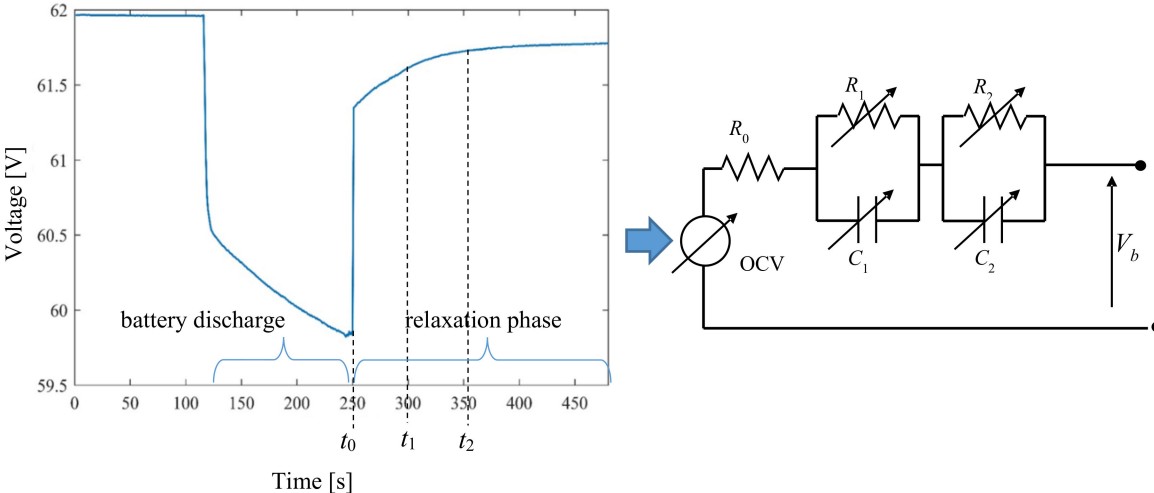

**Figure 4.** Battery voltage measurement (left) from which to derive the battery equivalent electrical model (right).

In fact, with reference to Figure 4, the battery voltage variation during the time interval from $t_0$ to $t_2$ can be represented by means of two *R–C* circuits in series. The voltage behaviour in $t_0$ can be modelled by means of a resistance $R_0$.

Hence, the circuit equivalent parameters can be obtained by solving the system below, consisting of the characteristic equations of the circuit shown in Figure 4, for different currents, SoCs and temperatures.

$$\begin{cases} R_0 = \frac{V(t_0) - V(t_1)}{I_i} \\ \frac{dV_{R_n C_n}}{dt} = \frac{I}{C_{n(SoC,\vartheta)}} - \frac{V_{R_n C_n}}{R_{n(SoC,\vartheta)} \cdot C_{n(SoC,\vartheta)}} \\ dV = E_1 (1 - e^{-\frac{t_1}{\tau_1}}) + E_2 (1 - e^{-\frac{t_2}{\tau_2}}) \\ \tau_n = C_n R_n \end{cases} \quad (1)$$

On the basis of the instantaneous values of current, SoC and temperature during the simulations, the correct battery parameters $R_0(SoCi,\vartheta)$, $C_n(SoC,i,\vartheta)$, $R_n(SoC,i,\vartheta)$ with $n = 1,2$, are automatically computed by means of advanced interpolation functions [19]. The instantaneous battery SoC during the simulation is computed as:

$$SoC\% = \frac{Q - \int_{t_0}^{t_i} i(t)dt}{Q} \cdot 100 \tag{2}$$

where $Q$ is the battery maximum capacity.

Hence, the instantaneous battery voltage $V_b$ can be computed as:

$$V_b = OCV(SoC) - R_0 \cdot I - V_{R_1C_1} - V_{R_2C_2} \tag{3}$$

where OCV is the battery open circuit voltage.

Figure 5 shows a screenshot of the battery Simulink model.

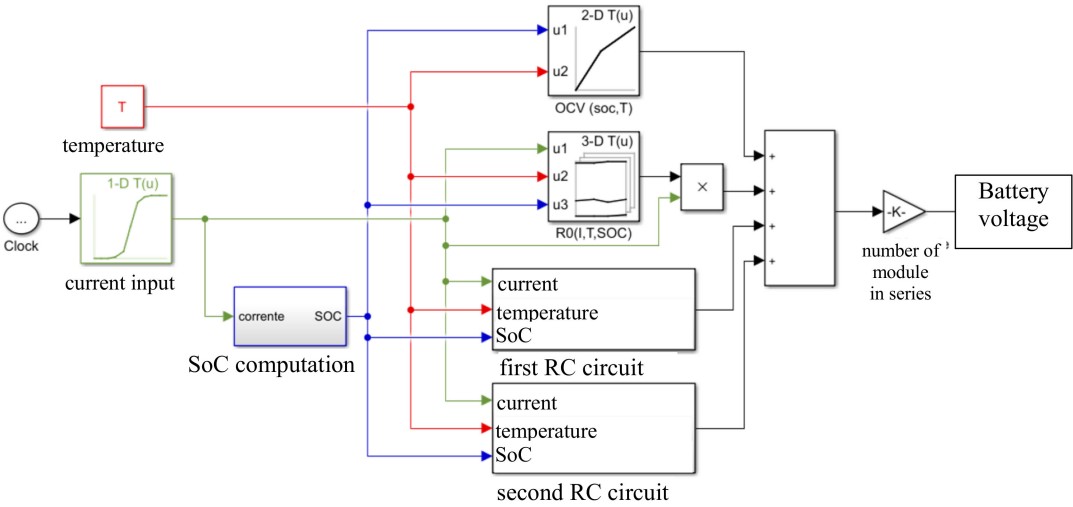

**Figure 5.** Battery model in the Simulink environment.

## 2.1. Battery Electric Model Validation

The battery model was tested by simulating all the performed aging cycles. The experimental current profiles of the aging tests were used as inputs for the model. Then, the output voltage of the model was compared with the experimental one. It is interesting to observe how the battery model perfectly follows the real battery voltage output for the initial tests, with an average error of 0.55%. In Figure 6, the 15th and 16th aging cycles are shown, and it is possible to note that in the simulation, the battery recharges take place at the same instant as in the experimental test.

By contrast, as the number of aging cycles increases, the moment when the battery needs to be recharged differs more and more between the experimental data and the simulation. Figure 7 shows a comparison between the simulation results and the experimental measurements for the 260th and 261th aging cycles. The difference between the two curves is due to the fact the battery model does not take into account the aging of the battery. As it is possible to see, the model error is visibly too high and not acceptable.

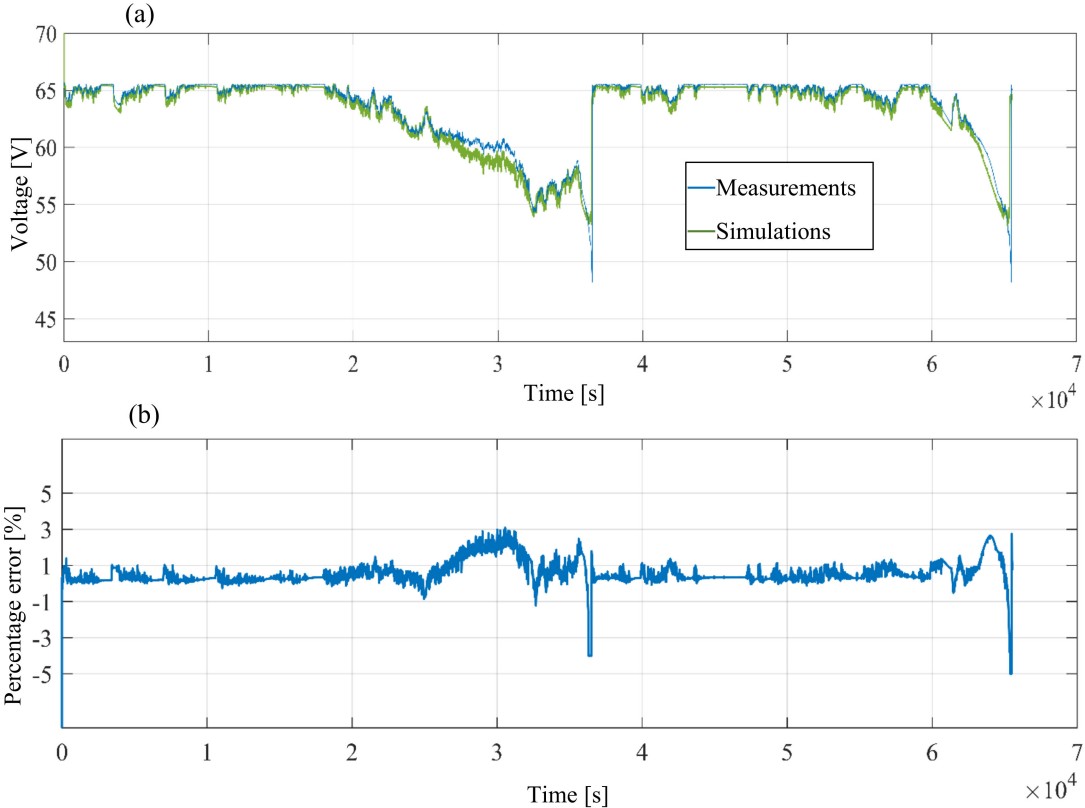

**Figure 6.** (**a**) Comparison between measurements and model results for the 15th and 16th aging cycles; (**b**) Model percentage error.

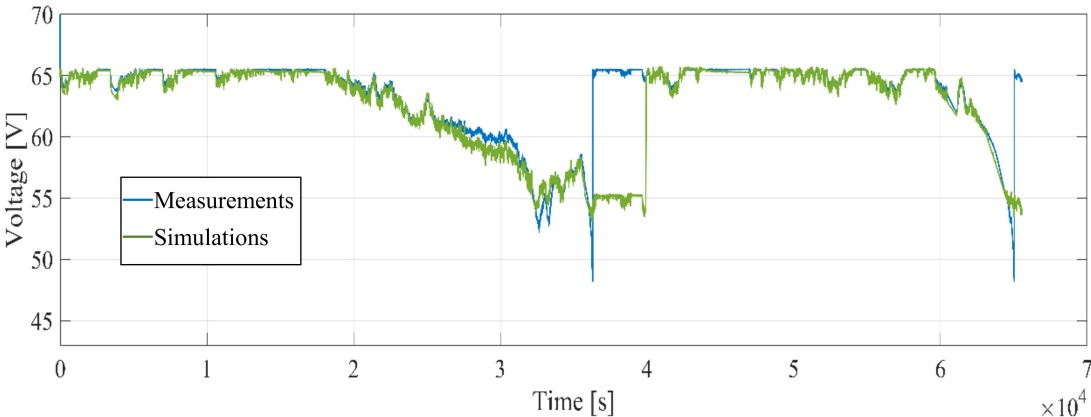

**Figure 7.** Comparison between measurements and model results for the 260th and 261th aging cycles.

In order to improve the model, the lost battery capacity $Q_{loss}$ was computed by means of the approach suggested in [22] by J. Wanga et al.:

$$Q_{loss} = Be^{\left(-\frac{Ea}{R\vartheta}\right)}Ah^Z \tag{4}$$

where:

$Ea$ is the battery activation energy [J/mol], $R$ is the ideal gas constant [J/(mol·K)] and $\vartheta$ the temperature [K].

The coefficients $B$ and $Z$ are derived by exploiting the "fittype" function of Matlab and using as inputs the experimental data of the aging tests.

The result of the fitting procedure is shown in Figure 8.

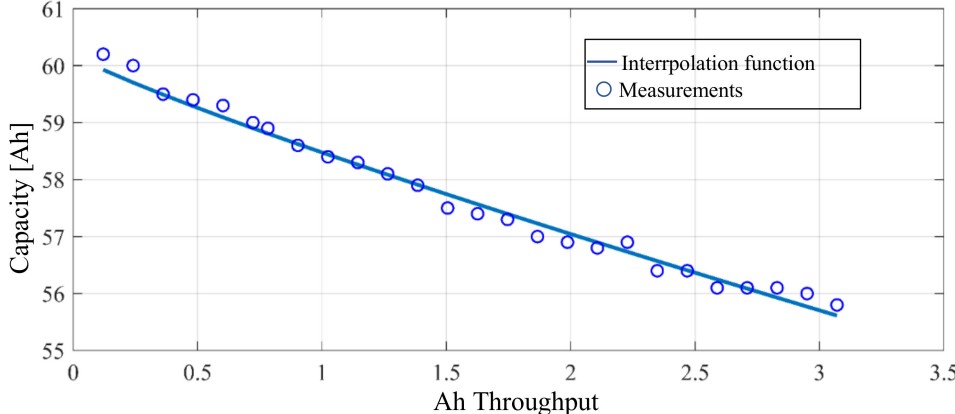

**Figure 8.** Comparison between the capacity interpolation function and the experimental measurements.

Once the $Q_{loss}$ computation has been implemented in the model, the battery SoC computation is updated by means of the following equation:

$$SoC\% = \frac{(Q - Q_{loss}) - \int_{t_0}^{t_i} i(t)dt}{(Q - Q_{loss})} \cdot 100 \tag{5}$$

The simulations fit very well with the experimental measurements for all the aging tests, with an average error lower than 1%. This precision is satisfactory in order to foresee the battery electrical behaviour during the grid frequency regulation operation. By way of example, Figure 9 shows the same comparison of Figure 7 by taking into account the battery aging effects.

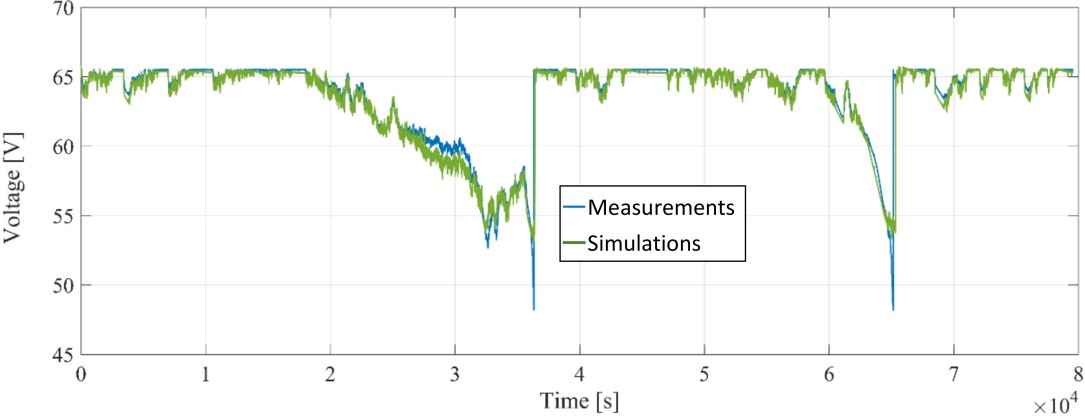

**Figure 9.** Comparison between model results and experimental measurements once the aging effect has been taken into account in the model.

It is worth noting that the implemented aging model cannot be considered as a general one for LMO cathode based batteries. Rather, it is specific for the test carried out in this particular research.

However, the result of the above presented analysis suggests that the battery aging effects caused by frequency regulation cycles could be related to the energy amount that has to be exchanged with the grid, which is proportional to the Ah throughput. Moreover, by taking into account the experimental evidence that emerged in the measurement campaign presented in [14], the frequency of the power supplied by the battery should be limited as well in order to counter the battery aging during frequency regulation cycles.

The basic idea behind this paper is to share the power that the storage system has to supply between batteries and a flywheel, by means of a suitable power management. Therefore, the second part of this work is the development of a precise flywheel electrical model.

## 3. Flywheel Electrical Model

### 3.1. Flywheel System

The flywheel system consists of a 20 kW-rated module based on a steel rotor coupled to a 4-poles brushless electrical machine (EM) with surface permanent magnets (PMs). The power rating complies with commercially available components; higher power requirement can be fulfilled by combining more units. Installing the EM outside the flywheel housing, usually kept at low pressure to reduce aerodynamical drag losses, simplifies both the EM–AC/DC power converter connection and the EM cooling [23]. A sketch of the flywheel system is provided in Figure 10, showing also the DC bus circuit shared with the battery modules.

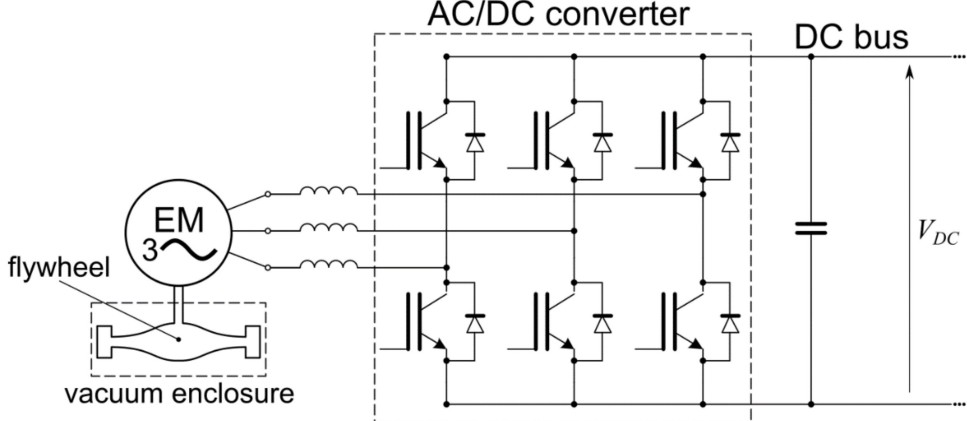

**Figure 10.** Sketch of the flywheel energy storage system.

The main system operational data and sizes are reported in Table 2. The speed and temperature range identify the minimum and maximum allowable values during the operating conditions. The maximum $\Omega_{max}$ and minimum $\Omega_{min}$ speed values are defined as a trade-off between deliverable energy per cycle and the electromechanical constraints which remarkably affect the system design. In fact, both the maximum allowable stress for the rotary components and the torque/voltage/current limits for the electromagnetic and electronic appliances must be considered. The maximum temperature is constrained by insulation and PM thermal limits. The flywheel rotor (FWR) is designed assuming an AISI 4340 steel (ultimate strength $\sigma_u = 1790$ MPa) truncated De Laval disc with an outer ballast [24]. The resulting shape and velocity factors $K = 0.96$ and $\chi = 2.51$ yield the mass values 69.3 kg and 3.7 kg for the disc and the ballast, respectively.

**Table 2.** Main operational FWR an EM data and sizes.

| Operational Data | | Sizes | |
| --- | --- | --- | --- |
| Rated power | 20 kW | Active length | 120 mm (EM) |
| Speed range | 10,833 rpm÷32,500 rpm | Airgap width | 2 mm (EM) |
| Max current rms value | 223 A | Outer diameter | 400 mm (EM) 500 mm (FWR) |
| Max torque (@40 °C) | 17.62 Nm | Flywheel disc height | 91 mm |
| Max phase voltage (@40 °C) | 90 V | Rotating mass | 9 kg (EM) 73 kg (FWR) |
| Temperature range | 25 °C ÷ 130 °C | Rotational inertia | 1.17 kgm² |

PM EMs are generally preferred for their higher efficiency and power density. Coreless stator configurations can be particularly convenient due to negligible no-load electromagnetic losses and low coil inductance, the latter aspect profitable the dynamic torque control promptness [25]. On the other side, they rely on high grade PMs and complicated winding assembly, so in this paper the more

conventional iron cored EM configuration is considered, benefiting from reduced ohmic losses during frequent charge/discharge operations—generally required for grid frequency regulation—and the chance to adopt ferrite PMs, with negligible eddy current losses and operating temperature (300 °C) remarkably higher than rare earth PMs, even if with lower magnetic performances. The magnetic properties at 40 °C (retentivity $B_r$ = 0.384 T, coercivity $H_c$ = 272 kA/m) fulfill the voltage and torque ratings at $\Omega_{max}$ and $\Omega_{min}$ respectively, provided that a 3-mm gap (including the airgap width and the PM retainment sleeve thickness) is held between the PM and the inner stator diameters.

The developed flywheel system model enables us to simulate the mechanical dynamics assuming the EM steady-state operation, by virtue of significantly smaller electrical time constants than mechanical ones and fast response of the electronic converter regulation system. However, most of the system parameters depend (often nonlinearly) on various process variables, such as temperature, speed and current, and therefore reliable models taking into account such phenomena are essential to effectively assess the power exchange and the charge/discharge time intervals. Many papers focus on the detailed evaluation of the losses in high speed brushless machines, covering both mechanical, fluid dynamic and electromagnetic contributions [26,27]. In this paper, the model predictive purpose has addressed the adoption of straightforward formulations which reproduce the loss dependency on the main physical parameters. Preliminary numerical simulations (i.e., transient finite element analyses) are, however, carried out to determine accurately the core loss components, which are quite critical for an iron-cored EM.

### 3.2. Formulation of the Model Parameters

The variation of the EM internal temperature $\theta$ significantly affects some EM parameters, and therefore the EM heating and cooling process during an operating cycle should be adequately modelled. For this purpose, a typical first order time evolution $\theta(t)$ is supposed, according to a suitable thermal time constant $\tau_{th} = R_{th}C_{th}$ with $R_{th}$ and $C_{th}$ the constant EM thermal resistance and capacitance, respectively. The parameter values are selected by examining commercial EMs with similar ratings provided with natural cooling. The $\theta(t)$ profile will be determined by the total EM losses $P_l(t)$ which are in turn affected by the temperature itself.

The EM parameters affected by $\theta(t)$ are the phase winding resistance $R_s$ and the PM retentivity $B_r$ in accordance with the PM temperature coefficient $k_{Br}$. Such dependence impacts on both the ohmic losses $P_j$ and the electromagnetic torque $T_{em}$ according to the expressions:

$$P_j = 3 \cdot R_s(\theta) \cdot I^2 = 3\, R_{s,40°}[1 + \alpha \cdot \Delta\theta_{40°}] \cdot I^2, \; T_{em} = k_T(\theta) \cdot I = k_{T,40°} \cdot \alpha_{Br}(\theta) \cdot I \tag{6}$$

with $\alpha$ the temperature coefficient of the copper resistance, $\Delta\theta_{40°} = \theta - 40$ °C, $k_T$ torque constant and $\alpha_{Br} = 1 - \frac{k_{Br}}{100}\Delta\theta_{40°}$. $R_{s,40°}$ and $k_{T,40°}$ correspond to the parameter values at the reference temperature $\theta^* = 40$ °C. Equation (1) apply if the voltage is promptly adjusted by the current control following to the PM e.m.f. and the resistance variations and if $k_T$ is not considerably affected by the armature current, as generally happens in surface mounted PM EMs. Being the dependence on current $I$ negligible, the core losses $P_{Fe}$ can be expressed as functions of speed $\Omega$ and temperature $\theta$ as:

$$P_{Fe} = \left[P^*_{hys,40°}\left(\frac{\Omega}{\Omega^*}\right) + P^*_{ec,40°}\left(\frac{\Omega}{\Omega^*}\right)^2\right] \cdot \alpha^2_{Br}(\theta) = b_{hys}(\theta) \cdot \Omega + b_{ec}(\theta) \cdot \Omega^2 \tag{7}$$

with $P^*_{hys,40°}$ and $P^*_{ec,40°}$ hysteresis and eddy current losses at the reference values $\Omega^* = \Omega_{max}$ and $\theta^*$. Equation (7) assumes that both the contributions are proportional to the square of the flux density.

The mechanical losses $P_m$ are related to the windage and friction resistances. According to the EM-FWR arrangement, the former is only related to the EM rotor; on the contrary the latter involves

the total suspended EM rotor–FWR mass $M_s$ and depends on the bearing type and geometry. An approximate $P_m$ evaluation can be obtained by the following expression:

$$P_m = P_w + P_{fr} = a_w\Omega^3 + a_{fr}\Omega \, , \ a_w \cong 0.3D_o^5 \cdot \left(1 + 5\frac{L_i}{D_o}\right) \cdot \left(\frac{60}{2\pi}\right)^3 \cdot 10^{-6} \, , \ a_{fr} = \mu_b M_s g \frac{d_b}{2}$$

where $D_o$ and $L_i$ are the rotor outer diameter and length, $d_b$ and $\mu_b$ are the bearing diameter and friction coefficient and $g$ is the gravitational acceleration. The semiempirical formulation of the coefficient $a_w$ is valid for an EM without a fan and when speed is higher than 15,000 rpm [25]. Both $a_w$ and $a_{fr}$ are assumed to be independent of temperature. By neglecting the power consumption of the vacuum pump, the total losses are therefore given by $P_l = P_j + P_{Fe} + P_m$.

### 3.3. Simulation Scheme

The flywheel system model is implemented in a Matlab/Simulink® code using the parameter values provided in Table 3. It is worth noticing that at $\Omega_{max}$ the mechanical losses have the most relevant contribution ($\approx$1530 W); on the contrary, the ohmic losses are the most important item at $\Omega_{min}$, ranging from about 435 W to 610 W according to the temperature value.

**Table 3.** Parameters of the flywheel system model.

| Quantity | Value | Quantity | Value |
|---|---|---|---|
| Phase resistance | $R_{s,40°} = 0.3$ m$\Omega$ | Thermal time constant | $\tau_{th} = 60$ min |
| Torque constant | $k_{T,40°} = 0.079$ Nm/A | Thermal capacitance | $C_{th} = 4\ 10^4$ J/°K |
| PM temperature coefficient | $k_{Br} = -0.2$ %/°C | Reference speed | $\Omega^* = 32{,}500$ rpm |
| Mechanical loss coefficients | $a_{fr} = 0.0272$ Ws/rad $a_w = 3.63\cdot10^{-8}$ Ws$^3$/rad$^3$ | Core loss coefficients | $b_{hys,40°} = 0.081$ Ws/rad $b_{ec,40°} = 1.3\cdot10^{-8}$ Ws$^2$/rad$^2$ |

Figure 11 shows the scheme implementing the mechanical and the thermal equations (n° eqq.), as well as the power loss models (colored blocks). The flywheel operation is governed by a saturated PID controller that tunes the current value according to the error between the reference and the output electric power. For a prompter system response, the required power sign is preemptively assigned to $T_{em}$ value. Finally, the calculation of the mechanical loss torque $T_m = P_m/\Omega$ enables the step-by-step integration of the mechanical equation $Jd\Omega/dt = -(T_{em} + T_m)$ given the initial speed $\Omega_0$. The block 'Speed check' controls prevents the flywheel exceeding the speed limits, excluding it from the power supply if overspeed occurs.

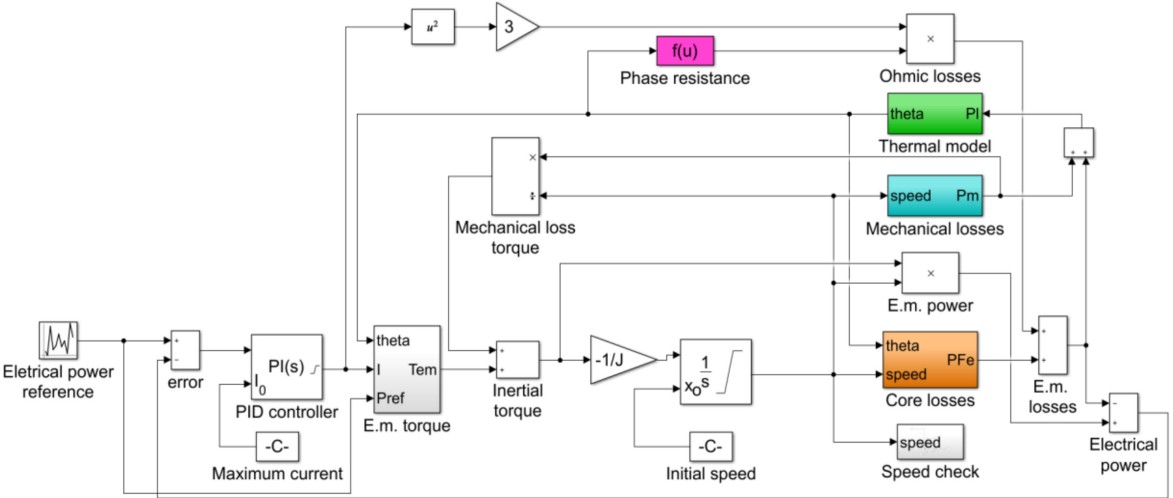

**Figure 11.** Simulink implementation of the flywheel system model.

Figure 12 reports some simulation results to check the proper flywheel operation as a stand-alone storage system. The power reference is intentionally created by a random function providing steep electrical power variations with both positive (generating) and negative (motoring) values. The power levels are within the flywheel rated power; the initial speed is fixed to $\Omega_0 = (\Omega_{min} + \Omega_{max})/2$. The curves evidence that:

- the power error is very low, confirming the current control effectiveness for the output electrical power regulation;
- the speed suitably tracks the requested accelerating and decelerating conditions; however, the prevalent output power requirement during this test results in a temporarily switch off of the flywheel system to recharge it;
- the temperature increase is more significant during the first part of the test because of the higher losses, particularly the mechanical ones; after that, it stabilizes to values compatible with short operating cycles.

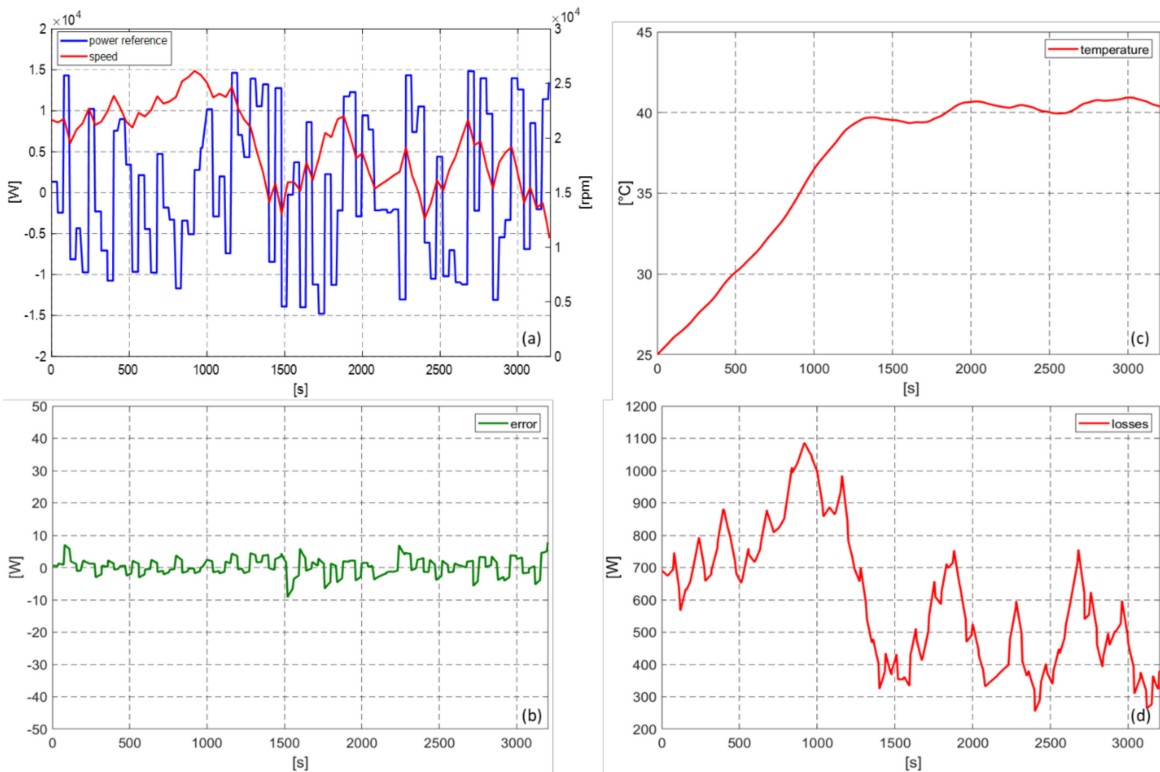

**Figure 12.** Simulation results related to the application of a test reference power; (**a**): output power reference and FWR speed; (**b**): error between reference and actual output power; (**c**): EM internal temperature; (**d**): EM losses.

As for the possible thermal issues, a temperature control can be implemented, which switches off the flywheel to prevent the temperature exceeding the maximum allowable value. It is worth pointing out that both $P_{Fe}$ and $P_m$ occur even under the idle condition; additional cooling could therefore be possibly required to lower more quickly the EM temperature to a satisfactory value.

## 4. The Hybrid Storage System Simulation

In this section, the above described battery and the flywheel models are joined to compose a hybrid storage system consisting of two 47.7 battery ranks (see Table 1) and a flywheel unit having the characteristics reported in Tables 2 and 3.

By means of a low-pass filter, the low frequency components of the required power are supplied by the batteries whereas the high frequency ones are supplied by the flywheel. Obviously, the lower the filter cut frequency, the lower the stress induced on the batteries. On the other hand, the filter cut frequency depends on the flywheel maximum power. For this hybrid system, the chosen filter cut frequency is equal to 1 mHz. When the high frequency components of the power exceed the maximum flywheel power, the battery supplies this gap. Figure 13 shows the Simulink scheme to manage the power between the two storage systems.

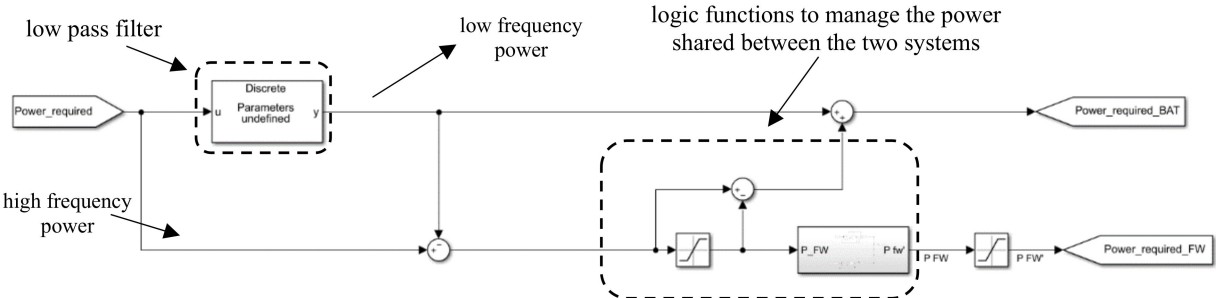

**Figure 13.** Power control of the hybrid storage system.

Figure 14 shows how the total amount of the required power is shared between the batteries and the flywheel.

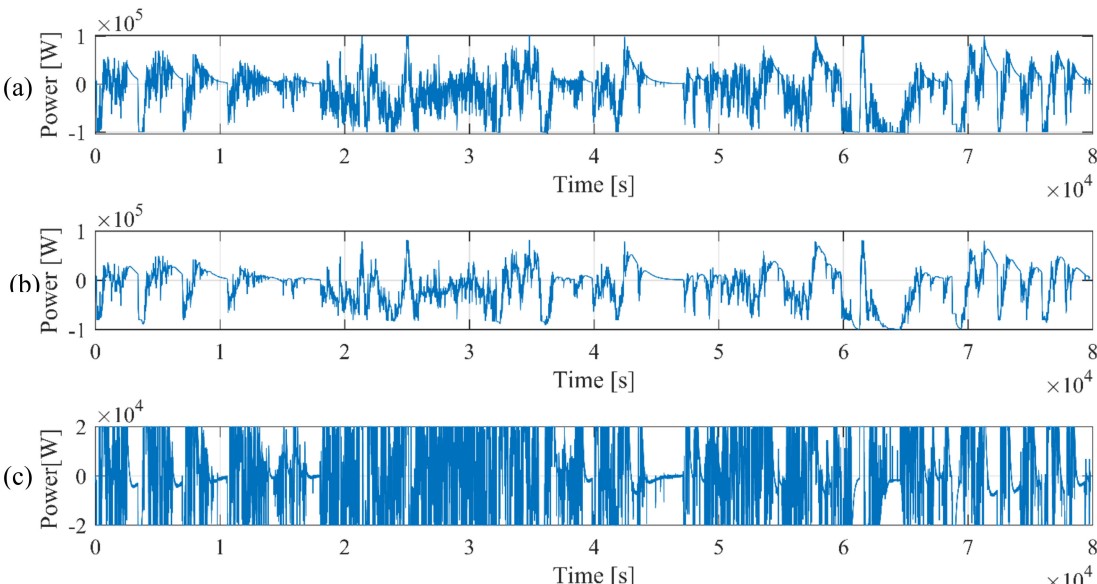

**Figure 14.** Hybrid system aging test simulation: (**a**) Total power; (**b**) battery power; (**c**) flywheel power.

In Figure 14b it is possible to see the effect of the power low-pass filter, which smooths the power that the battery has to supply.

It is worth considering that in this simulation, when the SoC of the flywheel is equal to zero, the flywheel recharges by absorbing power from the grid. In real grid frequency regulation operations, the flywheel should be rated so that its capacity is sufficient to support the primary frequency regulation, whose duration is about 15–25 s, before the flywheel SoC falls to zero. Only when the frequency of network returns within its tolerance band should the flywheel recharge.

By exploiting the developed models, 1424 accelerated aging cycles are simulated and the residual battery capacity for the battery model and for the hybrid one are compared. As it is possible to see in Figure 15, the battery model capacity at the end of the aging simulation is 9% less than the hybrid storage system one.

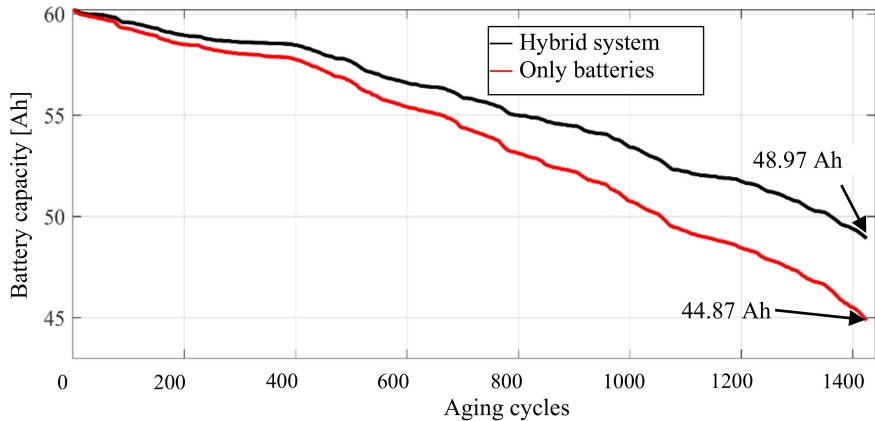

**Figure 15.** Comparison between the battery residual capacity after 1424 aging cycles for the two systems.

On the basis of the implemented models, it is interesting to determine the maximum number of aging cycles that each system can tolerate before the battery capacity decreases by 20%. For the considered module, the 20% decrease in the capacity corresponds to a residual capacity of 48 Ah. Figures 16 and 17 show the maximum number of cycles for the only-battery model and for the hybrid one, respectively. As it possible to see, in the hybrid system, the batteries can sustain 22.63% more aging cycles.

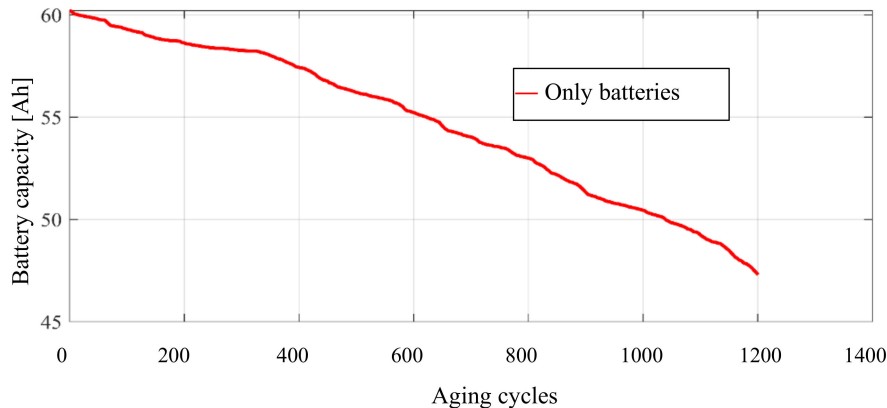

**Figure 16.** Only-battery storage system: maximum number of aging cycles.

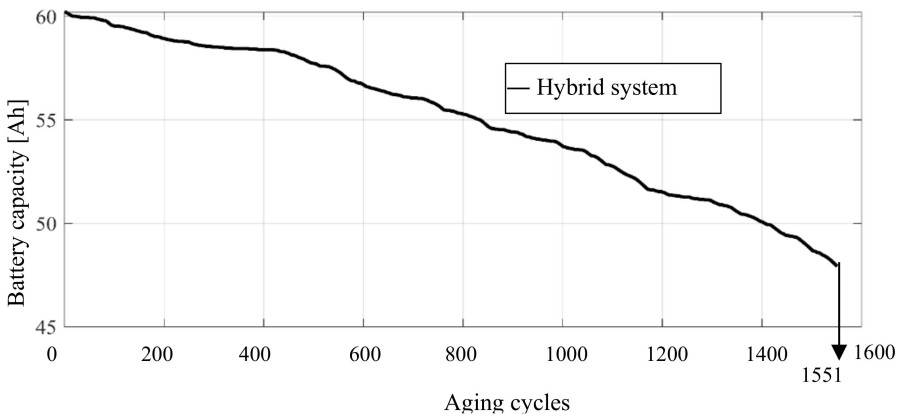

**Figure 17.** Hybrid storage system: maximum number of aging cycles.

## 5. Conclusions

In this paper, a model of a hybrid storage system, consisting of a battery rank supported by flywheel modules, is set up to analyze its operation for the grid frequency regulation. First, the experimental tests—generally difficult to retrieve—carried out on batteries with a LMO cathode under this operating condition enable us to tune a refined battery electrical model able to investigate battery aging. The good agreement between simulated and experimental results is encouraging, even if further experimental tests and research are necessary for a more accurate and general battery aging model. A refined flywheel-EM model is defined as well, taking into account the influence of both electrical and thermal quantities to accurately reproduce dynamic behavior. The flywheel-EM parameters are then identified to supply power fluctuations similar to the experimental tests.

The same tests are simulated by the integrated model of the hybrid storage system in the Simulink environment to define a control strategy, which properly allocates the high frequency component of the required power to the flywheel modules.

According to the simulation results, this power management applied to a two flywheel arrangement could reduce the aging of the tested batteries during frequency regulation cycles increasing the battery life by more than 20%. By such achievement, the reduction of battery modules over time for the same service can be assessed, as well as the resulting economic and environmental benefits.

**Author Contributions:** S.D.S. wrote and reviewed the paper, formally analyzed the experimental data, developed the battery electrical model, investigated the hybrid system behaviour and developed the hybrid system control. M.A. and A.T. wrote and reviewed the paper and developed the flywheel model. R.B. conceived the project, acquired the founding resources, wrote, reviewed and approved the final paper and supervised the entire work.

**Funding:** This work was financially supported by the Department of Industrial Engineering–University of Padova under the project entitled "Flywheels and Li-ion Batteries for Stationary Hybrid Energy Storage Systems".

**Conflicts of Interest:** The authors declare no conflict of interest.

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
