# Peer review of "Li-Ion Battery-Flywheel Hybrid Storage System: Countering Battery Aging During a Grid Frequency Regulation Service"

_applsci, doi:10.3390/app8112330_

Round 1
Reviewer 1 Report
The manuscript described a study of using flywheel to enhance the battery aging performance for grid frequency regulation service. The hybriding concept has been well illustrated. The simulation results show that a suitable control of the power shared two devices could effectively help countering the Li-ion battery accelerated aging due to the grid frequency regulation service. Please see the followings for some detailed comments.
1. About the name system of battery. In battery industry, it is not usually called LMO battery. It is just one type of Li-ion battery using LMO as cathode electrode.
2. Please label all the figures with a b c d with in one figure and give detailed figure captions.
3. In Fig 4a, the battery discharge process starts from 0 s or ~125 s? Why there’s no voltage drop in the first 125 s?
4. In table 2, please check speed and temperature range.
5. Please clarify the relationship between cycling time (s) and cycling cycles.
6. In Fig.12a, please separate rmp from W by put one Y scale to the right-hand side of the figure.
7. In Line 288, “By considering that a capacity decrease under 20% is usually considered the end of life of a battery”, please give supporting reference to this. The end of cycle life of battery depends on their chemistry and application. 20% of capacity decrease is usually considered for EV industry. The major concern is the resistance increase which require higher overpotential to charge the cell, which leads to the cathode decomposition within the voltage limit. In addition, lots of proposal is talking about the secondary use of EV battery in stationary energy storage.
Author Response
PREMISE: In the paper all the new text is in red colour
The authors wish to thank the reviewer for his fruitful comments, which in our opinion have improved the paper.
1. About the name system of battery. In battery industry, it is not usually called LMO battery. It is just one type of Li-ion battery using LMO as cathode electrode.
You are right. The used battery technology has been better defined in the revised version of the manuscript.
2. Please label all the figures with a b c d with in one figure and give detailed figure captions.
Done
3. In Fig 4a, the battery discharge process starts from 0 s or ~125 s? Why there’s no voltage drop in the first 125 s?
You are right, there was a mistake in the figure 4, in the first curly brace. The battery discharge starts at 125 s. A clarifying sentence has been added about this aspect. Thank you for your comment.
4. In table 2, please check speed and temperature range.
Done. The criteria behind the choice of such ranges are now specified.
5. Please clarify the relationship between cycling time (s) and cycling cycles.
The time duration of one cycle is equal to 10.2 h. This relation is reported in line 84.
6. In Fig.12a, please separate rmp from W by put one Y scale to the right-hand side of the figure.
Done
7. In Line 288, “By considering that a capacity decrease under 20% is usually considered the end of life of a battery”, please give supporting reference to this. The end of cycle life of battery depends on their chemistry and application. 20% of capacity decrease is usually considered for EV industry. The major concern is the resistance increase which require higher overpotential to charge the cell, which leads to the cathode decomposition within the voltage limit. In addition, lots of proposal is talking about the secondary use of EV battery in stationary energy storage.
You are right, the sentence has been changed in order to make it more correct.
Reviewer 2 Report
The paper presents a hybrid energy storage system with battery and flywheel for frequency regulation. The main idea is to use flywheel to reduce power requirement from battery, which in turn improves battery age. The paper deals with an important problem ans provides an interesting solution. Results indicate the potential for the proposed approach to to reduce battery aging. In my opinion, the paper can be accepted. However, I have a few comments that should be addressed.
- Literature review seems inadequate. The authors should do a thorough literature review and specifically state their contribution with respect to already existing works.
- Please mention the source of the current profile in Fig. 2.
- Many equations are not numbered. They should be numbered.
- In Fig. 6 and 7, the error plot should be given. Furthermore, the rms value of the error should be mentioned.
- In Fig. 13, how to design the low pass filter?
- What are the effects of uncertainties in aging model?
Author Response
PREMISE: In the paper all the new text is in red colour
The authors wish to thank the reviewer for his fruitful comments, which in our opinion have improved the paper, and for his appreciations.
- Literature review seems inadequate. The authors should do a thorough literature review and specifically state their contribution with respect to already existing works.
The reference section has been enriched.
- Please mention the source of the current profile in Fig. 2.
The current profile is related to the required power to perform the frequency regulation service. A clarifying sentence has been added in line 86.
- Many equations are not numbered. They should be numbered.
Done
- In Fig. 6 and 7, the error plot should be given. Furthermore, the rms value of the error should be mentioned.
You are right the error plot is helpful. The error plot has been implemented in the revised version of the paper in fig. 6 only for the sake of brevity, because in figure 7 the error between model and simulations is visibly too high and not acceptable. A sentence has been added to clarify this aspect and the average error has been calculated for the curves of fig. 6.
- In Fig. 13, how to design the low pass filter?
It has been simply used a Simulink low pass filter and the cut frequency has been set to 1 mHz.
- What are the effects of uncertainties in the aging model?
On the basis of the comparison between the simulation results and the experimental measurements, once the aging model was added in the electrical model, its precision was more than satisfactory if considering the scope of paper, with an average error lower than 1%. This error includes both the aging model and the battery electrical model uncertainties. A clarifying sentence has been added in revised version of the paper, in line 184.
Reviewer 3 Report
The manuscript should be published in Applied Sciences due to its importance. It deals with highly important topic. But before any publication i suggest following minor revision.
In this form introduction is too specific. Authors should bring the introduction part more general. This form it is too specific. For that purpose I highly suggest to authors to add some recent developments of supercapacitors which are also important in storage systems. Thus please see and cite following recent research: Nanoscale 10 (4), 1877-1884 (2018); Scientific reports 7 (1), 11222 (2017)
it will be good if the authors give numbers to each equations not some of them. Numbering starts at section 3.2
Why here LMO was used. For example why not Lithium Titanate? And what is cathode and what is anode then?
Accept after minor revision.
Author Response
PREMISE: In the paper all the new text is in red colour
The authors wish to thank the reviewers for his fruitful comments, which in our opinion have improved the paper, and for his appreciations.
The manuscript should be published in Applied Sciences due to its importance. It deals with highly important topic. But before any publication I suggest following minor revision.
In this form introduction is too specific. Authors should bring the introduction part more general. This form it is too specific. For that purpose I highly suggest to authors to add some recent developments of supercapacitors which are also important in storage systems.
-Thus please see and cite following recent research: Nanoscale 10 (4), 1877-1884 (2018); Scientific reports 7 (1), 11222 (2017)
We added a reference regarding the installation of supercapacitor for grid services which is more in tune with the scopes of the paper. Thank you for your comment.
-it will be good if the authors give numbers to each equations not some of them. Numbering starts at section 3.2
Done
-Why here LMO was used. For example why not Lithium Titanate? And what is cathode and what is anode then?
For this research we used LMO technology because it seems to be the better compromise between performance and costs. For example, Lithium Titanate technology has a better aging performance than the LMO, but it is much more expensive.You are right, the battery anode and cathode have to be cited in the introduction of the paper. Thank you for your comment.
Reviewer 4 Report
The manuscript submitted by a simulation study that demonstrates a power management system having a battery and a flywheel could boost the battery lifetime. The key to this lifetime enhancement is the distribution of the high-power energy portion to the flywheel.
The content of this manuscript is suitable for the submitted journal. The study is also meaningful as batteries are gaining increasing attention as energy storage components for modern energy management. This manuscript can be accepted once the following issues are cleared.
Major issues:
1. More technical details of the accelerating aging test (line 54) should be presented. For example, what are the charging and discharging rates?
2. Please number each equation for clear readability.
Minor issues:
1. Line 40: Please give some examples of the "some battery technologies".
2. Please provide a reference for the claim "...a capacity decrease under 20% is usually considered the end of life of a battery". Additionally, this sentence is incorrect. It should read as "...a capacity decrease of more than 20%...".
3. In the first part, the authors identified that the LMO battery aging correlated to the frequency regulation cycles and the frequency of the power supply. It will be great if the authors can comment the underlying reasons or mechanisms to account for the observed frequency-dependent aging phenomenon.
4. Figure 2: Why the number of cycle number was 500 (line 65)? Why was the cycle time set to be 10.2 hours each (line 66)?
Author Response
PREMISE: In the paper all the new text is in red colour
The authors wish to thank the reviewers for their fruitful comments, which in our opinion have improved the paper, and for their appreciations.
Major issues:
1. More technical details of the accelerating aging test (line 54) should be presented. For example, what are the charging and discharging rates?
You are right, the charge/discharge rate it important to better understand the aging tests. Unfortunately, it is not possible to identify a precise charge/discharging rate. The test has been performed by applying a variable current profile, with very fast transition from the charge to the discharge mode, and with very fast changes in current magnitude. A clarifying sentence has been added in the revised version of the manuscript about this issue in line 86.
2. Please number each equation for clear readability.
Done
Minor issues:
1. Line 40: Please give some examples of the "some battery technologies".
Done
2. Please provide a reference for the claim "...a capacity decrease under 20% is usually considered the end of life of a battery". Additionally, this sentence is incorrect. It should read as "...a capacity decrease of more than 20%...".
You are right. The sentence has been changed without referring to the battery end of life but just by considering a capacity decrease of 20% in the case study. Thank you for your comment.
3. In the first part, the authors identified that the LMO battery aging correlated to the frequency regulation cycles and the frequency of the power supply. It will be great if the authors can comment the underlying reasons or mechanisms to account for the observed frequency-dependent aging phenomenon.
You are right, it would be a great result. The battery aging dependence on power frequency has been hypothesized by observing the measurement campaign described in [14]. Unfortunately, this research has been carried out by exploiting electrical models which are not suitable for deeply explore the electrochemical reasons behind these phenomena but just foresee the battery electrical behaviour. Concerning this point, we are planning further experiments, trying to join researchers with different field of expertise.
4. Figure 2: Why the number of cycle number was 500 (line 65)? Why was the cycle time set to be 10.2 hours each (line 66)?
By applying a current profile which follows a typical grid power requirement during the frequency regulation service, the battery is discharged after 10.2 hours. The aging test were stopped after 500 cycles (about 7 months of experiments) because the data we have collected seemed sufficient to carry out a preliminary aging/electrical analysis, in order to understand how to improve the experimental analysis we are going to carry out in the near future.